# DASH: Decentralized CASH for Federated Learning

**Md Ibrahim Ibne Alam**
Department of ECSE
Rensselaer Polytechnic Institute
Troy, NY, USA - 12180
alamm2@rpi.edu

**Koushik Kar**
Department of ECSE
Rensselaer Polytechnic Institute
Troy, NY, USA - 12180
kark@rpi.edu

**Theodoros Salonidis**
IBM T.J. Watson Research Center
Yorktown Heights, NY, USA - 10598
tsaloni@us.ibm.com

**Horst Samulowitz**
IBM T.J. Watson Research Center
Yorktown Heights, NY, USA - 10598
samulowitz@us.ibm.com

## Abstract

We present DASH, a decentralized framework that addresses for the first time the Combined Algorithm Selection and HyperParameter Optimization (CASH) problem in Federated Learning (FL) settings. DASH generates a set of algorithm-hyper-parameter (Alg-HP) pairs using existing centralized HPO algorithms which are then evaluated by clients individually on their local datasets. The clients transmit to the server the loss functions and the server aggregates them in order to generate a loss signal that will aid the next Alg-HP pair selection. This approach avoids the communication complexity of performing client evaluations using communication-intensive FL training. FL training is only performed when the final Alg-HP pair is selected. Thus, DASH allows the use of sophisticated HPO algorithms at the FL server, while requiring clients to perform simpler model training and evaluation on their individual datasets than communication-intensive FL training. We provide a theoretical analysis of the loss rate attained by DASH as compared to a fully centralized solution (with access to all client datasets), and show that regret depends on the dissimilarity between the datasets of the clients, resulting from the FL restriction that client datasets remain private. Experimental studies on several datasets show that DASH performs favorably against several baselines and closely approximates centralized CASH performance.

## 1 Introduction

**Motivation.** Federated learning (FL) involves training a model from data located at client sites, without requiring clients to share their data. Recent studies Li et al. (2020a) have shown how a model can be trained optimally despite this sharing restriction through an iterative process of transmitting model parameters and aggregating them at a central server. However, this may incur substantial communication cost due to possibly many rounds of message exchanges of model parameters between server and clients. Much of the research on FL have focused on reducing this communication overhead during model training. However, such FL techniques typically assume a given algorithm and hyper-parameter (HP) setting, which must be agreed upon a priori by the clients, or be decided by the server and provided to the clients.

Towards developing AutoML capabilities in FL settings, recent work has shown that the training accuracy is highly dependent on the HPs and addressed the problem of HyperParameter Optimization (HPO) in FL settings (termed FL-HPO). In this paper we take a step further by addressing for the

Workshop on Federated Learning: Recent Advances and New Challenges, in Conjunction with NeurIPS 2022 (FL-NeurIPS'22). This workshop does not have official proceedings and this paper is non-archival.

first time the Combined Algorithm Selection and HPO (CASH) problem in the FL context, which we term FL-CASH. CASH is a central problem in AutoML systems and has been addressed in the centralized settings where all data is available in a central location, mostly by treating it as a more complex HPO problem that merges the HPs of all algorithms and adds the algorithm type as a new HP. This incurs an explosion in the problem dimensionality as different algorithms have different HPs. Extending FL-HPO algorithms to solve the CASH problem using this centralized approach would not be practical. Existing FL-HPO algorithms compute HPs locally and then aggregate them to a global HP set. The explosion on HP dimensionality would put a much higher burden to the local client HPO computations. Furthermore, the local client HPOs may yield CASH HPs of different type and it is not evident how to aggregate such HPs to a single optimal HP set.

**Overview of DASH.** We propose and evaluate *DASH*, a *D*ecentralized *CASH* framework for FL systems. DASH solves FL-CASH through an iterative process where algorithm and HP pair (Alg-HP) selections are computed at the central server and communicated to clients, and the clients generate a loss signal transmitted back to the server for the next Alg-HP selection. Such a loss signal could be generated using any known FL training process in the literature and then evaluating the resulting model on the client validation datasets. However FL training process incurs high communication overhead. In *DASH*, the clients perform evaluation of the Alg-HP selection made by the server, by training and validation in their local datasets, and transmit their loss values to the server. The server approximates the global loss for its current Alg-HP selection by aggregating the loss values. Thus, we use a two-level optimization approach where we perform HPO at server and local evaluations at the clients for each algorithm separately and finally select the (Alg-HP) pair that has the best loss. This avoids the use of the popular HPO expansion approach of centralized systems, allows the utilization of well established centralized HPO algorithms at the server, and averts the high message exchange complexity of FL model training by approximating the loss signal based on the losses of the clients' individual datasets. Thus, DASH follows a "single-shot" approach for global model training: the time and communication intensive global FL training needs to be performed only once, when a final algorithm and HP (Alg-HP) pair has been chosen by DASH. This implies that the total number of communication rounds required by DASH is proportional to the number of iterations required by the HPO process – which is fairly small, based on our evaluations.

We provide a theoretical analysis of DASH that bounds its sub-optimality in loss performance (regret) with respect to a centralized CASH solution that has access to all client data. This regret is expressed in terms of the dissimilarity across client datasets, which can be viewed as the performance penalty for decentralization. However, our evaluation on 7 large datasets with 7 algorithms shows that the performance of DASH closely approximates that of centralized CASH, and is significantly better when compared to three baseline approaches.

**Key contributions.** To summarize, the key novel contributions of this work are as follows.

- We present DASH, a framework for solving the CASH problem in a FL setting by performing algorithm selection and HPO at the central server, but the underlying model training and evaluation for such selection is done locally at the clients. This limits the communication complexity of DASH to a small number of rounds of (Alg-HP) pair message exchanges between the central server and the clients, which are a negligible cost compared to that of the FL training that is executed once the best (Alg-HP) pair is found.
- We provide a theoretical analysis of the worst-case loss performance of DASH, as compared to that of a centralized CASH solution that has access to all client datasets. The performance bound is expressed in terms of the dissimilarity between the client dataset distributions, capturing the impact of the FL restriction that the client datasets remain private.
- For 7 large data sets with 7 algorithm choices, we numerically compare the loss performance of DASH with a centralized CASH solution as well as two other baseline approaches. These baseline approaches include one that uses a default HP setting for each algorithm, and another that uses a state-of-the-art FL-HPO approach but with a fixed algorithm that performs well across most datasets.

## 2   Related work

While there is no previous FL-CASH approach, we briefly describe prior work on centralized CASH, FL model training, and FL-HPO approaches towards positioning DASH in this context.

*FL model training.* A number of optimization techniques have been devised to address the communication and computation overhead during FL training Li et al. (2020a). These techniques assume that algorithm and HPs are known, and can therefore be viewed as complementary to DASH; any of these approaches can be used for training the FL model once the Alg-HP pair has been chosen by DASH.

*Centralized CASH solutions.* A common approach is to view the CASH problem as an expanded HPO problem by merging the HPs of all algorithms and introducing the algorithm type as a new HP Komer et al. (2014), and then utilizing existing HPO methods (such as Shahriari et al. (2015); Li et al. (2017); Klein et al. (2017); Falkner et al. (2018). However, the heterogeneity of the HP spaces of different algorithms, along with the explosion of the HP space resulting from this approach can limit its efficiency. Some recent approaches have used reinforcement learning Efimova et al. (2017), adaptive allocation of HPO iterations to algorithms Li et al. (2020b), and alternating direction method of multipliers Liu et al. (2020). DASH enables usage of all these approaches in FL setting because it performs CASH at the server which typically has more processing capacity than the clients.

*FL-HPO approaches.* Most FL-HPO work focuses on finding local client HPs such as learning rates (Koskela and Honkela, 2019; Mostafa, 2019; Reddi et al., 2020), number of local SGD iterations (Wang et al., 2019) or network architecture parameters (He et al., 2020; Garg et al., 2020; Xu et al., 2020) for SGD training algorithms and deep neural networks (DNNs). At the central server, the local client HPs are then aggregated or extrapolated (through a regression based approach, for example Zhou et al. (2022)) to obtain a good HP setting for the entire global dataset. For the embedded FL-HPO problem in DASH, we take a different approach where the HPO problem is solved at the central server but by utilizing loss signals generated at the clients. This approach has the benefit of only requiring clients to run model training and evaluation on their local datasets instead of expensive FL training for finding the best global model, while allowing the use of well established centralized HPO solutions at the server.

## 3 FL-CASH Problem Formulation

Similar to the standard CASH problem considered in a centralized setting Thornton et al. (2013); Zöller and Huber (2021), in the FL-CASH problem we are given a set of algorithms $\mathcal{A} = (A^{(1)}, \cdots, A^{(J)})$, where each algorithm $A^{(j)}$ is associated with hyperparameters (HPs) that belong to domain $\Lambda^{(j)}$. Each algorithm choice $A^{(j)}$ and HP setting $\boldsymbol{\lambda} \in \Lambda^{(j)}$, compactly written as $A_{\boldsymbol{\lambda}}^{(j)}$, is associated with a model class $\mathcal{W}_{\boldsymbol{\lambda}}^{(j)}$, from which a model (parameter vector) $\boldsymbol{w} \in \mathcal{W}_{\boldsymbol{\lambda}}^{(j)}$ must be chosen so as to minimize a predictive loss function $\mathcal{L}(\boldsymbol{w}, \mathcal{D}')$ over a validation dataset $\mathcal{D}'$.

In an FL setting McMahan et al. (2017); Yang et al. (2019), the training dataset $\mathcal{D}$ is partitioned into several subsets $\mathcal{D}_i, i \in \mathcal{C}$ that are owned individually by a set of $N = |\mathcal{C}|$ clients. Thus $\mathcal{D} = \cup_{i \in \mathcal{C}} \mathcal{D}_i$. We assume that $\mathcal{D}_i$ is private to client $i$, and cannot be shared or aggregated due to privacy or complexity reasons. The training and validation parts of the dataset $\mathcal{D}_i$ are denoted by $\mathcal{D}_i^{\text{train}}$ and $\mathcal{D}_i^{\text{valid}}$, respectively; therefore $\mathcal{D}_i = \mathcal{D}_i^{\text{train}} \cup \mathcal{D}_i^{\text{valid}}$. Given algorithm choice $A^{(j)}$ and HP choice $\boldsymbol{\lambda}$, written compactly as $A_{\boldsymbol{\lambda}}^{(j)}$, an FL algorithm $\mathcal{F}$ aims to determine a model $\boldsymbol{w}$ using the training dataset,

$$\mathcal{F}(A_{\boldsymbol{\lambda}}^{(j)}, \cup_i \mathcal{D}_i^{\text{train}}) \longrightarrow \boldsymbol{w} \in \mathcal{W}_{\boldsymbol{\lambda}}^{(j)},$$

where the training dataset is written as $\cup_i \mathcal{D}_i^{\text{train}}$ to emphasize its distributed (partitioned) nature. Usually, $\boldsymbol{w}$ is chosen to minimize the training error, i.e., the FL algorithm $\mathcal{F}$ typically aims to minimize $\mathcal{L}(\boldsymbol{w}, \cup_i \mathcal{D}_i^{\text{train}})$ over $\boldsymbol{w} \in \mathcal{W}_{\boldsymbol{\lambda}}^{(j)}$, using iterative methods that involve local model training at the individual clients (using their private datasets) and sharing information on models and their accuracies (but not data) with a central aggregator. Lets us denote $\mathcal{D}^{\text{train}} = \cup_i \mathcal{D}_i^{\text{train}}$ and $\mathcal{D}^{\text{valid}} = \cup_i \mathcal{D}_i^{\text{valid}}$. Then given the underlying FL algorithm $\mathcal{F}$ for finding the model (for a chosen Alg-HP setting), the CASH problem for FL involves finding $A_{\boldsymbol{\lambda}^\star}^\star$ that minimizes a global loss function, expressed as the aggregation of loss functions at the clients computed over their validation datasets

$$A_{\boldsymbol{\lambda}^\star}^\star = \underset{\substack{A^{(j)} \in \mathcal{A}, \\ \boldsymbol{\lambda} \in \Lambda^{(j)}}}{\arg\min} \mathcal{L}(\mathcal{F}(A_{\boldsymbol{\lambda}}^{(j)}, \mathcal{D}^{\text{train}}), \mathcal{D}^{\text{valid}}) = \underset{\substack{A^{(j)} \in \mathcal{A}, \\ \boldsymbol{\lambda} \in \Lambda^{(j)}}}{\arg\min} \sum_i \alpha_i \mathcal{L}(\mathcal{F}(A_{\boldsymbol{\lambda}}^{(j)}, \cup_i \mathcal{D}_i^{\text{train}}), \mathcal{D}_i^{\text{valid}}), \quad (1)$$

where $\alpha_i$ are appropriately defined client weights, such as $\alpha_i = \frac{1}{N}$ or $\alpha_i = \frac{|\mathcal{D}_i|}{|\mathcal{D}|}$.

---
**Algorithm 1** *DASH*: *D*ecentralized *CASH* for Federated Learning
---

1: **Input:** $\mathcal{A}$, $\Lambda^{(j)}\ \forall j$.
2: $\tilde{\mathcal{A}}$= [Set of all algorithms].
3: **for** each algorithm $A^{(j)} \in \tilde{\mathcal{A}}$ **do**
4:     Set the initial HP $\boldsymbol{\lambda}^{(j)}(1)$ to a default or random HP in $\Lambda^{(j)}$.
5:     **for** $k = 1$ to $HPO_{iter}$ **do**
6:         Central server sends $\boldsymbol{\lambda}^{(j)}(k)$ to all clients
7:         Each client $i$ trains model with HP $\boldsymbol{\lambda}^{(j)}(k)$ and reports the loss $l_i(k)$.
8:         Server calculates the aggregated loss in that iteration as $l(k) = \sum_i \alpha_i l_i(k)$.
9:         Based on the aggregate loss values calculated so far, $l(k'), k' \le k$, server sets new HP,
            $\boldsymbol{\lambda}^{(j)}(k + 1)$, by executing the next step of a given HPO algorithm.
10:     **end for**
11:     Store the best (HP, loss) for each algorithm.
12: **end for**
13: Set $A^{\dagger}$ = algorithm with the best loss after the last HPO iteration; $\boldsymbol{\lambda}^{\dagger}$ = corresponding HP choice.
14: For chosen Alg-HP pair $A^{\dagger}_{\boldsymbol{\lambda}^{\dagger}}$, compute the best global model $\boldsymbol{w}^{\dagger}$ using any given FL algorithm.
15: **Output:** $A^{\dagger}, \lambda^{\dagger}, \boldsymbol{w}^{\dagger}$.

---

## 4 Algorithm and Analysis

**DASH-Algorithm.**    While solving this FL-CASH problem, we need to satisfy the FL requirement that the datasets $\mathcal{D}_i$ remain private, while attempting minimize the number of communication rounds between the server and the clients, including the rounds needed by the federated learning (model training) algorithm $\mathcal{F}$. The development of DASH, outlined in Algorithm 1, is guided by these practical requirements.

DASH approximates the global loss for any $A^{(j)}_{\boldsymbol{\lambda}}$ by aggregating each of the client's losses (computed on their individual datasets), i.e.,

$$\sum_i \alpha_i \mathcal{L}(\mathcal{F}(A^{(j)}_{\boldsymbol{\lambda}}, \mathcal{D}^{\text{train}}_i), \mathcal{D}^{\text{valid}}_i). \tag{2}$$

Comparing with (1)), it can be noted that $\cup_i \mathcal{D}^{\text{train}}_i$ in the argument of the model training function $\mathcal{F}$ in (1) is replaced by $\mathcal{D}^{\text{train}}_i$ in (2). This implies that in *DASH*, model training (and therefore loss computation) happens locally in each client, avoiding the communication-intensive procedure of computing the global model through FL. This allows *DASH* to compute the global loss function (albeit approximately), for a given $A^{(j)}_{\boldsymbol{\lambda}}$ and training dataset, in a single round of communication (see appendix for communication overhead calculation). The DASH algorithm comprises of three key elments, which are described below.

*Decentralized HPO:* The decentralized HPO method works as follows, which is repeated until the maximum number of iterations ($HPO_{iter}$) is reached. The aggregator (central server) sends out an initial HP setting (say a default or randomly chosen HP) to all clients. Upon receiving this HP setting, the clients use their private datasets to train the model and report back the respective performances (i.e., losses or accuracies). The server then calculates the global loss value for that HP setting by aggregating all per-client loss values. The next HP setting is then decided based on the global loss values, using any centralized HPO technique chosen by the server. See steps 5-10 of Algorithm 1.

*Algorithm Selection: DASH* performs decentralized HPO (as described above) for all the algorithms and picks the algorithm (and its best HP setting) that obtains the best global loss value by simply comparing those values at the central server. See step 13 of Algorithm 1.

*Federated Model Training:* After the Alg-HP pair has been decided (as described above), *DASH* computes the model with a single run of FL training, for which any FL algorithm can be used.

**Regret Analysis.**    In this section, we provide a theoretical analysis of the loss optimality of *DASH*, given by Theorem 1 below; proof of the result is included in the Appendix as supplementary material. For ease of exposition, we are going to use $A \in (A^{(1)}, \cdots, A^{(J)})$ to denote a generic algorithm, and $A_{\boldsymbol{\lambda}}$ to denote a generic Alg-HP pair.

| DataSet | Default | FLoRA-LGBM | CASH-C | DASH |
|---|---|---|---|---|
| Connect-4 | 72.97 | 74.24 | 75.54 | 74.77 |
| Default of.. | 75.41 | 75.50 | 75.51 | 75.49 |
| Diabetic .. | 53.36 | 55.81 | 56.79 | 56.32 |
| EEG-Eye | 93.10 | 92.79 | 94.04 | 93.82 |
| Electricity | 91.04 | 90.32 | 93.51 | 93.51 |
| Higgs | 72.18 | 72.12 | 72.56 | 72.27 |
| Magic Tel.. | 86.13 | 86.65 | 86.66 | 86.40 |

Table 1: Comparison of *Default*, *FLoRA-LGBM*, *CASH-C*, and *DASH*

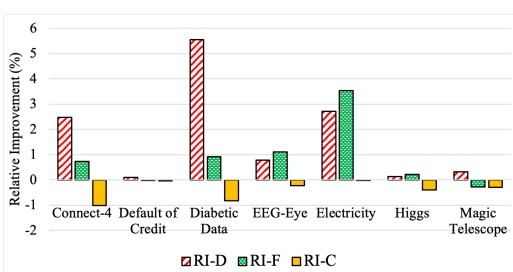

Figure 1: $RI$ of *DASH* on Baselines

Table 2: Effect of $N$

| DataSet | $RI - D$ | | $RI - F$ | |
|---|---|---|---|---|
| | N = 5 | N = 20 | N = 5 | N = 20 |
| Connect-4 | 2.56 | 2.75 | 0.81 | 0.99 |
| Default of.. | 0.09 | -0.08 | -0.03 | -0.21 |
| Diabetic .. | 5.29 | 5.43 | 0.66 | 0.79 |
| EEG-Eye | 0.82 | 0.52 | 1.16 | 0.85 |
| Electricity | 2.64 | 2.02 | 3.46 | 2.84 |
| Higgs | 0.05 | -0.53 | 0.12 | -0.448 |
| Magic Tel.. | 0.33 | 0.20 | -0.27 | -0.39 |

We first define $\hat{\mathcal{L}}(A_{\boldsymbol{\lambda}}, \hat{\mathcal{D}})$, the loss rate of FL on any given dataset $\hat{\mathcal{D}}$ for a chosen Alg-HP pair $A_{\boldsymbol{\lambda}}$, as

$$\hat{\mathcal{L}}(A_{\boldsymbol{\lambda}}, \hat{\mathcal{D}}) = \mathcal{L}(\mathcal{F}(A_{\boldsymbol{\lambda}}, \hat{\mathcal{D}}^{\text{train}}), \hat{\mathcal{D}}^{\text{valid}}). \tag{3}$$

Then for any two given datasets (or their distributions) $\hat{\mathcal{D}}_1$ and $\hat{\mathcal{D}}_2$, and $\nu(\hat{\mathcal{D}}_1, \hat{\mathcal{D}}_2)$ being the 1-Wasserstein distance between them, we assume that

$$|\hat{\mathcal{L}}(A_{\boldsymbol{\lambda}}, \hat{\mathcal{D}}_1) - \hat{\mathcal{L}}(A_{\boldsymbol{\lambda}}, \hat{\mathcal{D}}_2)| \leq \beta' \nu(\hat{\mathcal{D}}_1, \hat{\mathcal{D}}_2), \tag{4}$$

holds for some constant (scalar) $\beta'$, across all Alg-HP pairs $A_{\boldsymbol{\lambda}}$.

**Theorem 1.** *The loss attained by DASH is at most* $[\beta' \sum_i \alpha_i \cdot \nu(\mathcal{D}_i, \mathcal{D})]$ *more than the optimum (centralized) loss.*

Theorem 1 implies that the sub-optimality of DASH depends on the dissimilarity between the client datasets; further, DASH is performance-optimal when all client datasets have the same distribution.

## 5 Empirical Evaluation

**Datasets.** To evaluate the performance of *DASH*, datasets from OpenML Vanschoren et al. (2013) were used. Initially 43 datasets were chosen such that each of them had instances within a range of 10,000 to 120,000. To check their performance consistency under different conditions, we chose 7 Algorithms and their corresponding HP spaces: *Random Forest, Decision Tree, Extra Tree, Logistic Regression, XGB, LGBM* and *MLP*. To reduce computational complexity, the HP space chosen for each of these 7 Algorithms covers a subset containing their most important HPs (see appendix). Upon checking the performance (accuracy) of all the 43 datasets for different Algorithms with default HPs and centralized HPO, we chose 7 datasets (see Table 1 dataset column) for performance evaluation of (*DASH*). The reason to choose these datasets were; i) they produce consistent results (for accuracy) over many runs with different data-seeds and cross-validations; ii) they are diverse in terms of number of examples and number of features.

**Baselines.** Since there is no existing FL-CASH solution, DASH is compared with 3 baselines. The first, called *Default*, is when there is no HPO and the default sci-kit learn HP configuration of the Algorithms are used. These default HPs have been empirically shown to yield good performance across many datasets. The second baseline called *FLoRA-LGBM*, runs FLoRA Zhou et al. (2022), a recent single-shot FL-HPO algorithm; the Algorithm in this case is fixed to LGBM, which often yields best performance across different datasets. The third baseline, called *CASH-C* is the CASH solution assuming all client data is centralized.

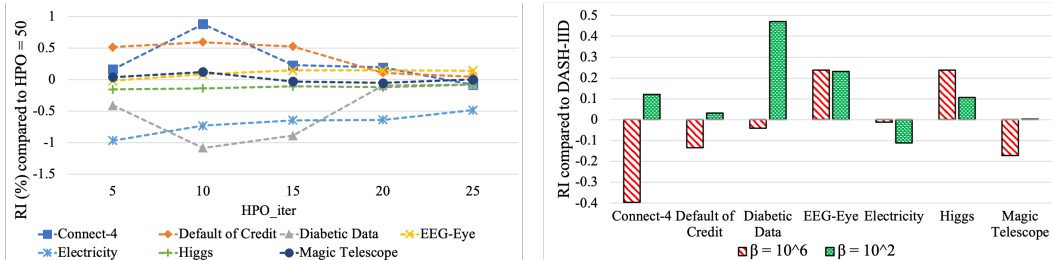

Figure 2: Effect of $HPO_{iter}$          Figure 3: Effect of $\beta$

**Implementation.** *DASH* was implemented by broadly following Algorithm 1. To check consistency of results, 25 random seeds were used to split data to clients and perform HPO (i.e., using hyperopt) After the generation of clients data, each client's data are split in 70-30 ratio to use as train and test data. We evaluate Alg-HP configurations at the clients on the training dataset using cross-validattion with CV = 10 number of folds. The test data is not used during *DASH* algorithm Alg-HP search. It is only used to evaluate the performance of the best Alg-HP configuration on the model computed using the FL training on this configuration. In order to check the consistency of the results we measured the standard deviation across the 25 random seeds. To perform HPO for each training algorithm at the server, we used hyperopt Komer et al. (2014), however, any other HPO techniques could have been used. In most of our runs $HPO_{iter} = 50$ was used (except otherwise specified).

**Comparing with Baselines.** To compare the performance of *DASH* with the baselines outlined above, all datasets were divided into three similar sized segments (using different random seeds each run), and each of those segments represented a client for the *DASH* algorithm. We define Relative Improvement ($RI$) of *DASH* over any baseline ($B$) as, $RI - B = \frac{P_D - P_B}{P_B} \times 100\%$, where $P_D$ is the performance (accuracy) of *DASH* and $P_B$ is the performance of any of the baselines. To denote Relative Improvement ($RI$) over *Default*, *FLoRA-LGBM* and *CASH-C* we use the terms $RI - D, RI - F$ and $RI - C$ respectively. The value of these three relative improvements are depicted in Fig. 1, and the actual performances for all the three baselines along with *DASH* are given in Table 1. As expected, *DASH* outperforms both the *Default* and *FLoRA-LGBM* in most cases, and gives a performance quite close to the (centralized) optimal solution (*CASH-C*).

**Effect of Client Number.** Table 2 summarizes the performance of DASH when the number of clients $N$ is 5 and 20. We note that most of the $RI$ values are positive (indicating better performance in *DASH*), and the few remaining negative values are quite small in magnitude (indicating similar performance from *DASH*). Moreover, for some datasets (i.e., Diabetic Data, Connect-4), *DASH* shows a high rate of performance improvement.

**Effect of HPO Iteration.** Fig. 2 shows the effect on the performance of *DASH* when $HPO_{iter}$ is changed. We define the $RI$ metric for this case as, $RI - HPO = [(P_{iter} - P_D)/P_D] \times 100\%$, where $P_D$ is the performance of *DASH* with $HPO_{iter} = 50$ (the maximum number of HPO iterations we used) and $P_{iter}$ is the performance achieved by the given $HPO_{iter}$. We ran the experiments five times (using different seeds) for each $HPO_{iter} = 5, 10, 15, 20$ and $25$; and took the average performance. Smaller $HPO_{iter}$ yielded higher variation in the results, and in some cases the performance is even 0.5 to 1% better than the performance of $HPO_{iter} = 50$. One major reason behind this performance anomaly is due to different data distribution of train and test data, where higher $HPO_{iter}$ can lead to over-fitting issue in some cases. At $HPO_{iter} = 25$, such variations decreased and most of the performances are similar to the performance of $HPO_{iter} = 50$.

**Effect of non-IID Clients.** In all of the simulations till now, clients are generated by dividing the whole dataset using some random data-seed. Due to large amount of instances, we can assume these data distributions of each client to be similar (IID). We call the performance of *DASH* with this random data distribution as *DASH*-IID. To create different distribution of the target variable at different clients, we use a Dirichlet constant $\beta$ (smaller $\beta$ yields more heterogeneous non-iid distribution). The $RI$ values of different values of $\beta$ ($10^2$ and $10^6$) over *DASH*-IID are depicted in Fig. 3. We observe that RI has a small range ($-0.4\%$ to $0.5\%$) for both values of beta across all datasets which shows that DASH achieves stable performance on heterogeneous non-iid distributions.

# 6 Conclusion

We propose *DASH* as a decentralized method to solve CASH problem for FL platform. *DASH* performs algorithm selection and HPO at the central server via a few rounds of communication with the clients. The communication between the server and client is limited to the exchange of Alg-HP pairs chosen by the server and the corresponding loss values calculated by the clients. The worst case sub-optimality of *DASH* in terms of loss performance was analyzed theoretically, and found to be proportional to the dissimilarity in the clients' dataset distributions. Empirical evaluations show that *DASH* performs very close to the optimum solution (CASH-C) and generally outperforms the other two baselines discussed. Consistent results over 7 large datasets and different settings (e.g., number of clients, different data distribution of clients) demonstrates the strong potential of *DASH* for being a effective solution to the FL-CASH problem.

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

## A   Supplementary Material

**Communication Overhead of DASH:**   If $L_{HP}$ ($L_{\text{loss}}$) denotes the maximum length of the message needed to communicate an HP setting (loss value, resp.) from the server to a client (from a client to the server, resp.), then the communication overhead of the server for executing the CASH process in DASH (which is all due to running the decentralized HPO) is bounded by $N \times (L_{HP} + L_{\text{loss}}) \times HPO_{iter}$. Our evaluations suggest that a reasonably small value of $HPO_{iter}$ (say about 20) suffices to get near-optimal loss performance in practice; this implies that total message exchange complexity of DASH can be expected to be small, unless the number of clients $N$ is very large. In addition to this, there is the complexity of doing one run of FL training process at the last step of DASH (see step 14 of Algorithm 1); this additional complexity will be minimally necessary in any solution of the FL-CASH problem, and will depend partly on the exact FL algorithm used in this step.

**Proof of Theorem 1:**   To prove Theorem 1, we first introduce some definitions. On given dataset $\hat{\mathcal{D}} = \cup_i \hat{\mathcal{D}}_i$, we define the optimum (centralized) loss as follows (see (1) and (3)),

$$\tilde{\ell}(A_{\boldsymbol{\lambda}}, \hat{\mathcal{D}}) = \sum_i \alpha_i \, \mathcal{L}(\mathcal{F}(A_{\boldsymbol{\lambda}}, \cup_i \hat{\mathcal{D}}_i^{\text{train}}), \hat{\mathcal{D}}_i^{\text{valid}}) = \mathcal{L}(\mathcal{F}(A_{\boldsymbol{\lambda}}, \hat{\mathcal{D}}^{\text{train}}), \hat{\mathcal{D}}^{\text{valid}}) = \hat{\mathcal{L}}(A_{\boldsymbol{\lambda}}, \hat{\mathcal{D}}). \quad (5)$$

Next, based on (2) and (3), we define the loss computed by DASH on the same dataset as

$$\ell(A_{\boldsymbol{\lambda}}, \hat{\mathcal{D}}) = \sum_i \alpha_i \, \mathcal{L}(\mathcal{F}(A_{\boldsymbol{\lambda}}, \hat{\mathcal{D}}_i^{\text{train}}), \hat{\mathcal{D}}_i^{\text{valid}}) = \sum_i \alpha_i \, \hat{\mathcal{L}}(A_{\boldsymbol{\lambda}}, \hat{\mathcal{D}}_i). \quad (6)$$

We first bound the difference between $\tilde{\ell}(A_{\boldsymbol{\lambda}}, \mathcal{D})$ and $\ell(A_{\lambda}, \mathcal{D})$, as follows.

$$
\begin{aligned}
& |\ell(A_{\boldsymbol{\lambda}}, \mathcal{D}) - \tilde{\ell}(A_{\boldsymbol{\lambda}}, \mathcal{D})| \\
&= |\sum_i \alpha_i \, \hat{\mathcal{L}}(A_{\boldsymbol{\lambda}}, \hat{\mathcal{D}}_i) - \hat{\mathcal{L}}(A_{\boldsymbol{\lambda}}, \hat{\mathcal{D}})| \qquad \text{[using (5) and (6)]} \\
&= |\sum_i \alpha_i \left( \hat{\mathcal{L}}(A_{\boldsymbol{\lambda}}, \hat{\mathcal{D}}_i) - \hat{\mathcal{L}}(A_{\boldsymbol{\lambda}}, \hat{\mathcal{D}}) \right)| \qquad \text{[since } \sum_i \alpha_i = 1] \\
&\leq \beta' \sum_i \alpha_i \nu(\mathcal{D}_i, \mathcal{D}). \qquad\qquad \text{[using (4)]} \qquad\qquad (7)
\end{aligned}
$$

Now, let us denote $A^\dagger_{\boldsymbol{\lambda}^\dagger}$ and $A^\star_{\boldsymbol{\lambda}^\star}$ as the Alg-HP pair chosen by *DASH* and the optimum (centralized) solution, respectively. Since $A^\dagger_{\boldsymbol{\lambda}^\dagger}$ and $A^\star_{\boldsymbol{\lambda}^\star}$ are chosen by optimizing using the loss functions $\ell$ and $\tilde{\ell}$, respectively,

$$A^\star_{\boldsymbol{\lambda}^\star} = \arg\min_{A_{\boldsymbol{\lambda}}} \tilde{\ell}(A_{\boldsymbol{\lambda}}, \mathcal{D}), \quad \text{and} \quad A^\dagger_{\boldsymbol{\lambda}^\dagger} = \arg\min_{A_{\boldsymbol{\lambda}}} \ell(A_{\boldsymbol{\lambda}}, \mathcal{D}).$$

In the above, we implicitly made the idealistic assumption that the central server is able to find an exact solution to the underlying HPO problem (but with loss function $\ell$) for any algorithm $A$ when implementing *DASH*. [In general, the correctness of the HPO solution will depend on the HPO algorithm used by the server, and the number of HPO iterations used, $HPO_{iter}$. If the HPO is solved inexactly, the general line of our analysis still holds, but an extra error term (that depends on the degree of approximation of the HPO solution) would need to be introduced in the regret bound.]

Now, we finalize our proof as follows,

$$\begin{aligned}
\ell(A^\dagger_{\boldsymbol{\lambda}^\dagger}, \mathcal{D}) &\le \ell(A^\star_{\boldsymbol{\lambda}^\star}, \mathcal{D}) && \text{[since } A^\dagger_{\boldsymbol{\lambda}^\dagger} \text{ minimizes } \ell \text{ ]} \\
&\le \tilde{\ell}(A^\star_{\boldsymbol{\lambda}^\star}, \mathcal{D}) + \beta' \sum_i \alpha_i \nu(\mathcal{D}_i, \mathcal{D}), && \text{[using Eq. 7]}
\end{aligned} \tag{8}$$

which completes the proof of Theorem 1.

**HP space used:** The HP spaces used for HPO in the *DASH* algorithm are as follows;

*RandomForest:*
'n_estimators' : uniformint(50, 300),
'max_depth' : uniformint(4, 20),
'min_samples_split' : uniformint(2, 6),
'min_samples_leaf' : uniformint(1, 3).

*DecisionTree:*
'max_depth' : uniformint(4, 20),
'min_samples_split' : uniformint(2, 6),
'min_samples_leaf' : uniformint(1, 3).

*ExtraTree:*
'max_depth' : uniformint(4, 20),
'min_samples_split' : uniformint(2, 6),
'min_samples_leaf' : uniformint(1, 3).

*LogisticRegression:*
'tol' : uniform(1e-4, 1e-3),
'C': uniform(0.2, 1.0),
'max_iter' : uniformint(80, 100).

*XGB:*
'eta' : uniform(0.1, 0.3),
'min_child_weight': uniformint(1, 3),
'max_depth' :hp.uniformint(3, 12).

*LGBM:*
'subsample' : uniform(0.5, 1.0),
'colsample_bytree': uniform(0.5, 1.0),
'max_depth' : uniformint(3, 12),
'min_data_in_leaf': uniformint(20, 30) ,
'num_leaves' : uniformint(20, 80).

*MLP:*
'hidden_layer_sizes' : uniformint(50, 200),
'alpha': loguniform(-5*np.log(10), 1*np.log(10)),
'learning_rate_init' : loguniform(5*np.log(10), -1*np.log(10)).

