# OpenReview forum: "DASH: Decentralized CASH for Federated Learning"
_NeurIPS.cc/2022/Workshop/Federated_Learning — FL-NeurIPS 2022 Poster_

### Official Review · Reviewer_xe6q · 2022-10-13

The paper proposes a decentralized hyper-parameter optimization and algorithm selection method for the federated learning framework. Specifically, because of the high dimension and communication cost of the centralized version, the paper proposes to sent the parameters copies to every clients and aggregate them in the optimized way in the server end. Experiments have been conducted on different datasets show the proposed method could improve significantly over the default hyperparamter setting and achieve comparable performance with the centralized CASH method.

Pros:
1. It is the first paper to study CASH problem in the federated learning setting.
2. The proposed method shows a good results on several datasets.

Cons:
1. The paper lacks the discussion of previously proposed method on hyperparameter tunning in Fed learning.  As the paper is the first paper to study CASH inFed ML, it should clearly state the difference between HPO and CASH in FedML. For example, it is necessary to mention why HPO is applicable but CASH is not?
2. It is not clearly shown why the centralized CASH method is not directly applicable.
3. The baseline is kind of weak. It should include some HPO methods in FedML as the baselines by justing keeping the algorithm selection as the same.

---

### Official Review · Reviewer_u9v5 · 2022-10-15
**The authors introduced a decentralized framework to addresses the FL-CASH problem**

The authors introduced a decentralized framework that addresses the FL-CASH problem. Here, the review has the following concerns:
1. It would be better to introduce the motivation of using CASH more.
2. It would be better to provide a detailed literature review on FL-CASH.
3. Why (4) is a reasonable assumption?
4. Theorem 1 does not prove the convergence of designed algorithm.

---

### Official Review · Reviewer_bgmU · 2022-10-18
**The claim is interesting but the demonstration is not very convincing**

This paper introduces an algorithm for combined algorithm selection and hyperparameter optimization (CASH) in the federated setup. The proposed method is called Decentralized Combined Algorithm Selection (DASH).

DASH is a meta-algorithm. It can employ any existing Hyper-Parameter Optimization (HPO) method. The formulation of DASH is somewhat trivial: it loops over all possible algorithm selections, employs the HPO for all these algorithms separately, and then chooses the algorithm hyper-parameter combination that works the best across all attempts. Once the algorithm-parameter pair is set, DASH employs a (any) federated learning for the actual training.
The main novelty of DASH compared to the existing federated HPO approaches* is the following: DASH uses the server to do HPOs, and the role of clients is to train the model with provided algorithm-hyperparameters pair, evaluate the validation loss, and communicate this loss to the server. So the mathematical problem tackled in DASH is in fact different than the problem solved in the centralized CASH problem.
The theoretical analysis is quite limited. The theorem says that the difference in the performance between the centralized solution and DASH can be bounded above by using the Wasserstein distance between the distributions; this bound comes simply from the triangle inequality.

The paper has a good discussion of past centralized CASH and HPO-FL approaches. The authors do a good job of comparing DASH against other approaches like (1) no hyperparameter tuning, (2) a case of client-based hyper-parameter generation, (3) a centralized CASH solution. They also investigate other relevant factors like the number of clients and the number of HPO iterations. It is not discussed thoroughly in the experiments, however, how does the method perform when the data distributions are non-IID.

The limitations of the work are not discussed, and no future research directions are suggested. The draft seems to be lacking in detail in some important places. The authors do a good job of putting forward details about the hyperparameter spaces chosen for each algorithm. I am concerned about the explanation of how the authors chose 7 datasets that they used in the experiments out of 43 in the benchmark. They say that the performance of the algorithms was consistent for these 7 datasets over random seeds. Does this mean that it was inconsistent for the rest?

The results are interesting because they suggest the one-shot server-based HPO where clients only consider their local datasets in the CASH problem can be competitive against the central CASH solutions in certain settings. The demonstration however is not fully convincing for the reasons mentioned above. Since the loss function for the complete data is simply the sum of the loss functions of the clients, the proposed method is simply training machine learning algorithms on different datasets after a sophisticated HPO algorithm to find hyperparameters that work moderately for all the data, which limits the originality of the work.

More questions:
Why was a similar search over the algorithms for DASH not done for FLoRA?
What is the centralized CASH solution that was used as a baseline and what is the rationale for choosing it?
What is the reason for not choosing any other HPO algorithm other than hyperopt?
Could you elaborate more on the assumption in (4)? Is this a common assumption in CASH problem literature?

My general conclusion is that this is a work in progress and needs significant improvements before it gets ready for publication.

For the workshop, I think this is a borderline submission. My main concern is that the analysis and numerics are not convincing enough. But as preliminary results, they still suggest that a one-shot approach might work moderately well for the CASH problem in FL. I am leaning slightly toward rejecting, but I will not get upset if the paper gets accepted.

---

### Decision · Program_Chairs · 2022-10-20

Accept (Poster)